# Analysis of Eco-Innovations in Peruvian Accommodation Establishments

**Alicia Lencia Cántaro Márquez, Greysi Fiorela Miranda Vásquez and Daysy Ángeles Barrantes ***

Facultad de Administración Hotelera, Turismo y Gastronomía, Universidad San Ignacio de Loyola, Lima 15024, Peru; alicia.cantaro@usil.pe (A.L.C.M.); greysi.miranda@usil.pe (G.F.M.V.)
* Correspondence: daysy.angeles@usil.pe; Tel.: +51-986-484-231

**Abstract:** Technical environmental innovations (TEIs), also called eco-innovations, are key tools within environmental management systems owing to their potential contribution in reducing environmental degradation. Thus, the importance of the use of TEIs in the hospitality industry was discussed in this study. A qualitative-method approach was used. To that end, six hotel managers from Peruvian accommodation establishments were interviewed. It was found that the use of environmental process innovations reduces operating costs and environmental discharges. However, management indicators are required to measure the real impact on the environment. Despite the benefits of TEIs, their use has not yet become widespread in the Peruvian hospitality industry owing to internal barriers, such as the company's size and investment capacity, and external barriers, such as the absence of a policy framework established by public management entities and the lack of suppliers.

**Keywords:** eco-innovation; environmental process innovation; environmental product innovation; hospitality industry; sustainability; environmental management





## 1. Introduction

Innovation implies far-reaching substantial changes in the sustainable development debate and the fields of environmental policy and management. Therefore, any innovation that significantly reduces environmental damage can be referred to as eco-innovation [1,2]. In the hospitality industry, technical environmental innovations (TEIs) may help to improve the sustainability of companies while enabling them to gain competitive advantages [3]. In this regard, benefits such as reduction in operating costs and optimization of the use of resources have been highlighted by recent studies [4,5]. Additionally, tourists are becoming increasingly aware of environmental care, especially in the aftermath of the COVID-19 pandemic; thus, sustainable hotels attract greater attention [6,7]. Because of this, green hotel investment is being supported to revitalize the industry [8]. At the international level, TEIs are already an essential part of the value proposals of important hotel companies [9].

In Latin America, only the use of end-of-pipe technologies predominates since they entail fewer developmental barriers and lower costs [10–12]. However, green innovations such as green products and clean technologies are far from being leveraged to their full extent [12–15]. Some factors that contribute to this are attributed to internal aspects of the business itself, ranging from a lack of competencies in hotel environmental management to a hotel's financial situation. At the same time, external pressures, in the form of actors such as policy-makers, suppliers, and consumers, to mention but a few, may ease or deter the uptake of eco-innovations [16].

Considering the scarcity of information on TEIs in the hospitality field, researchers tend to study the phenomenon broadly; usually, eco-innovations are included under the umbrella term of environmental practices. As a result, important characteristics of TEIs are overlooked. Conversely, the main objective of this research was exclusively to analyze the importance of the use of TEIs within the Peruvian hospitality context.

## 2. Literature Review

Following the publication of the Brundtland report, sustainable development became part of the political and business agenda [17]. At the business level, the balanced growth in economic, social, and environmental capital makes up the basis of the triple bottom line application of sustainability. In other words, a sustainable business management system must ensure that a company's social, environmental, and economic performance contributes to sustainable development. For this purpose, all sorts of innovations are needed, whether in the processes, the product, or the business system itself [18].

### 2.1. From Innovation to Environmental Innovation

In academia, innovation is crucial in the creation of solutions. Since the concept of eco-innovation was introduced, this notion has evolved to include aspects other than technology, such as the capacity to add value and diminish environmental risks [19–22]. Bossink [23] describes TEIs as initiatives to maintain, enhance, and renew the environmental and social quality of business services, products, and processes. It should be stressed that environmental performance and the degree of novelty, at least at the business level, are distinctive features of green innovations [21]. In other words, innovations need only be novel for the company, not necessarily the market.

Several studies highlight the existence of limitations and incentives that are internal, external, or linked to the inherent nature of eco-innovations, such as a country's regulations or market demand [2,24–26].

Based on the OECD [27] and Rennings et al. [28], this research took three categories of environmental innovations as its reference: product, process, and organizational environmental innovations. The latter were excluded from the present study. Organizational innovations are of a more complex nature, as they are related to structural changes such as workplace organization and even going as far as external relations [29], which would require a study of their own to be properly addressed.

Environmental process innovations are new or improved processes different from those previously adopted by the company, using fewer resources to produce goods and services [21,28,30]. In turn, they are subdivided into end-of-pipe technologies (EOP) and clean technologies (CT). EOPs are solutions to facilitate the elimination of pollutant discharges, either right before or after the end of the production process. LED bulbs are the most common example because they support energy-efficient lighting and are perceived to contribute to significant cost savings [13,31]. Meanwhile, CTs are preventive and integrated technologies that eliminate pollutants in the production cycle and reduce the use of raw materials [14,31–33]. Renewable energy technologies, such as solar panels, are included in this category because they can contribute to the reduction in carbon emissions [34,35].

Environmental product innovations encompass new or improved products that significantly differ from previously used ones [28]. They involve changes in aspects of the product's eco-design, functionality, and performance—the materials, design, ease of use, packaging, etc. [29,31,36]. Biodegradable items, such as toiletries, are a clear example of environmental product innovations [3].

### 2.2. Eco-Innovation in the Hospitality Industry

Hotels are considered to be among the commercial establishments that generate the most GHG due to the intensive consumption of resources required to operate 24 h a day [37,38]. Generally, energy and water resources represent the majority of consumption and costs in hotels; therefore, TEIs are most commonly used for their control and savings [13,34,39,40]. This explains why they are seen as generators of long-term economic and environmental benefits [41–43]. Other reasons that encourage the adoption of TEIs are compliance with environmental legislation and the creation of a positive public image [44–46]. For example, guests show high levels of satisfaction and loyalty to those establishments with a strong environmental commitment and that promote various green practices such as the use of TEIs [47]. In particular, there is a greater inclination to book rooms in hotels

that have eco-technologies [48]. However, in the hospitality industry there are low levels of diffusion of environmental innovations [49,50]. In addition, internal factors, such as the budget and the size of the organization, affect the TEI adoption process [3,13,51]. Hence, in Latin America use of TEIs is still incipient [52]. The following Table 1 clusters the set of works presented in the above lines.

**Table 1.** Fields of study within the framework of eco-innovations in the hospitality industry.

| Areas of Study | Sources |
|---|---|
| Economic aspects | Chan et al. [13]; Cuadros and Luza [34]; Pez and Torres [39]; Sánchez-Ollero et al. [40]; Samper et al. [45]; Velázquez et al. [46]; Asadi et al. [53]. |
| Environmental aspects | Chan et al. [13]; Cuadros and Luza [34]; Jaramillo et al. [37]; Wang et al. [38]; Elzek et al. [42]; Urbina [43]; Samper et al. [45]; Asadi et al. [53]. |
| Social aspects | Pez and Torres [39]; Samper et al. [45]; Merli et al. [47]; Memarzadeh and Anand [48]; Asadi et al. [53]. |
| Determining factors (motivations, barriers, incentives) | de Oliveira and da Cunha [3]; Chan et al. [13]; Deraman et al. [44]; Velázquez et al. [46]; Fichter and Clausen [49]; Kumar and Sheoran [50]; Magadán and Rivas [51]; Asadi et al. [53]. |

## 3. Methodology

### 3.1. Research Setting and Participant Sampling

Sustainability remains an understudied topic in the lodging industry [54]. Some limitations of the current literature, such as the scarcity of information on governmental regulations linked to TEIs in the hospitality industry, need to be addressed [54]. Additionally, as mentioned earlier, only a few studies about Peruvian accommodation establishments have focused on environmental innovations. This study adopted a qualitative approach to provide in-depth insights into the current state of the use of TEIs and its importance for hotels located in Peru [55,56].

In a qualitative inquiry, samples tend to be small, and participants are selected in a purposive manner [57]. For this study, the researchers looked for participants that had knowledge or experience regarding TEIs in their hotels. An additional pre-selection criterion, the level of authority, was included to determine potential participants. Thus, the sample consisted of owners, general managers, and operations managers. From the initial 15 hotels that were contacted, only 6 agreed to take part in the interviews. We reached saturation after the fifth interview.

It is important to note that all six lodging establishments were hotels located in the three main regions of Peru, the Andes, the Amazon, and the Coast. According to Peruvian regulations, the current classification of lodging establishments only distinguishes four types: hotels, aparthotels, hostels, and *hostales* [58]. Even though some of the participants indicated that the businesses they represented were ecolodges, this does not necessarily correspond to the current regulations. Additionally, these establishments used the term ecolodge for commercial purposes, specifically to convey that the hotel was environmentally friendly or located in a rural area.

### 3.2. Data Collection

An interview guide composed of 27 questions was used to collect non-numerical data through individual interviews, lasting approximately 1 h each. The questions were designed to understand the hotel managers' experiences with adopting TEIs and their opinions on how these eco-innovations added value to their businesses. These were structured interviews, where the participants answered the questions giving their honest opinions on the subject in question. All six interviewees gave their informed consent to participation in this study. Interviews were conducted between March and June 2022.

### 3.3. Data Analysis

After collecting the data, interview transcriptions were carried out. To respect each participant's anonymity, their names were coded as E1–E6. The transcriptions were checked for language errors using an online tool named Stilus.

Finally, a thematic analysis of the information was carried out through a categorization matrix and a coding technique to process the results and raise the relevant conclusions and recommendations. Since the number of interviews was small, data analysis was completed by manual coding. The results were grouped according to the type of eco-innovation, either environmental product innovations or environmental process innovations, and they were analyzed according to four thematic axes: economic aspects, environmental aspects, social aspects, and determining factors for the adoption of TEIs. This study employed the investigator-triangulation technique to ensure academic rigor. This triangulation type allowed for different perspectives to be included in making observations and drawing conclusions which enhanced the process of interpretation and analysis [55].

## 4. Results

### 4.1. Most Used TEIs

Regarding environmental process innovations, the most used type of CT was in the sewage treatment system through the implementation of biodigesters and biofilters.

"Every bathroom, even those that can be found in common areas like the restaurant, are connected to biodigesters" (E2)

"We have 7 biodigesters which we use as part of our human waste management system" (E4)

Similarly, at least one type of EOP technology was implemented. In this regard, it was found that the most used EOP innovations were devices that help reduce water consumption, such as aerators and water-saving filters "installed in showers and faucets" (E3).

Likewise, the most used environmental product innovations were biodegradable bathroom products.

"We offer biodegradable toiletries like shampoo and hand soaps, even the packaging is biodegradable" (E1)

"Restrooms are supplied with bio degradable products for hand and body washing" (E6)

However, there was little use of product innovations in cleaning and laundry areas. Compared to environmental process innovations, product innovations are limited to three types: amenities, kitchen utensils, and cleaning supplies. The results are summarized in Figure 1.

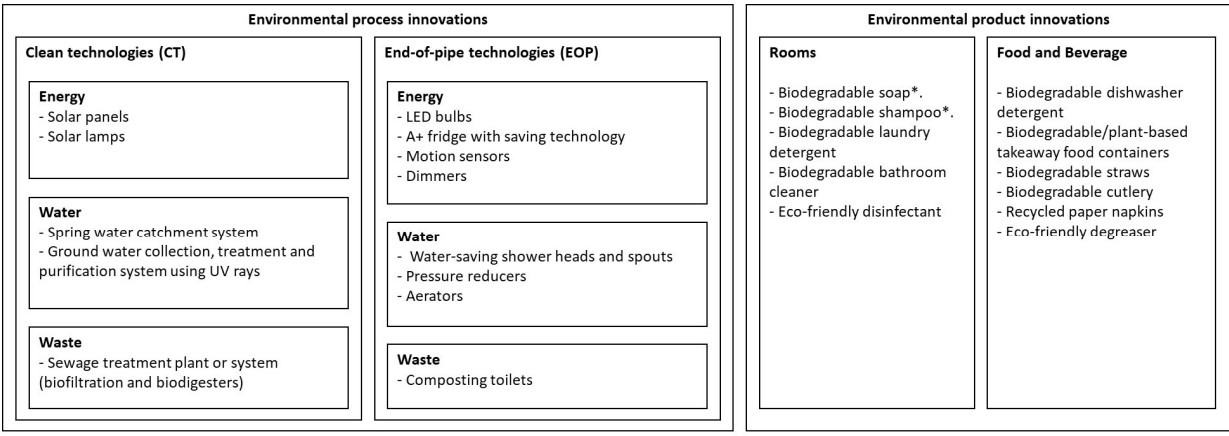

**Figure 1.** Environmental process and product innovations adopted by hotels.

*4.2. Environmental Aspects*

Utilizing environmental process innovations contributes to the reduction in waste and potable water consumption, especially through the reuse of effluents for irrigation of green areas, since "the solid part stays in the biodigester (...) and the liquid part goes under the garden, which means that we no longer use potable water" (E4). However, the reduction in environmental burdens or resources resulting from the use of process innovations is not usually quantified.

> "In the beginning, (...) we had to use a fuel-powered generator (...) [Now] we have reduced the 100% of the noise (...) with solar energy. Another thing (...) is the carbon released by the generator. (...) if an X amount was released, an amount that we do not have, now we know it is no longer released (...). I would not be able to compare (...) how much we have reduced (...)" (E1)

> "According to the manufacturer's specifications, these filters could generate around 30% in water savings" (E3)

> "We have reduced the use of electricity with dimmers (...) when you do not walk through an area, it will lower the light intensity, which results in lower consumption" (E5)

According to the interviewees, it is difficult to quantify the reduction in environmental burdens through the use of environmental process innovations, unless it is a radical change. In this regard, there is a need to carry out comprehensive monitoring of environmental process innovations to determine the reduction in the ecological footprint.

On the other hand, most of the remnants of environmental product innovations are organic, compostable, or biodegradable, and are easily processed in biodigesters which has helped reduce waste. In addition, logistical strategies and greater negotiating power with the suppliers of these products were considered as part of the overall reduction strategy these hotels had in place.

> "For example, soaps (...) were brought in small plastic envelopes. [Then] they changed them for paper packages (...), we decided to remove them and now we buy the soaps in boxes" (E4)

> "It is a logistical issue; (...) it is not the same thing to be sold several small bottles of something as it is to buy (...) by the gallon" (E2)

It was also found that the consumption of eco-friendly products takes the formula of the ingredients as a reference point. Other eco-design features (packaging, packing, and wrapping) are not taken into consideration to help reduce waste in the last stage of the products' life cycles (the final disposal). Furthermore, participants shared they did not check "as thoroughly" (E4) when purchasing these eco-products. Therefore, they could not guarantee that the product innovations being used, which claimed to be eco-friendly, have been developed or manufactured following sustainable processes, such as efficient consumption and transformation of raw materials.

*4.3. Economic Aspects*

Regarding environmental process innovations, for some participants it was difficult to determine a decrease in costs due to the private nature of certain resources and the eco-innovations themselves.

> "I could not give you an exact figure because we collect the water from springs in the area" (E3)

> "We used to pay 12 thousand soles a month; with the underground well (...) we pay 100 or 200 soles" (E5)

> "The water is our own, so we do not make a direct payment for a service (...); there is no expense, other than a minimal monthly maintenance" (E1)

Environmental process innovations replace conventional technologies for the provision of basic services. Despite this, there is no accurate record of consumption expenditure, so most interviewees assume that "you lower costs per se" (E4). Only maintenance and other minimal costs were considered relevant and attributable to the use of these TEIs.

"After 10 or 20 years, with proper maintenance, costs become extremely cheaper" (E2)

"Since we now have LED lights, kilowatts consumption decreases (...), which allows us to be a little more competitive due to this strategy that we have developed" (E4)

It is evident that several TEIs were implemented bearing in mind that the ROI would be observed in the long term. They have even helped the hotels' operational reduction strategy.

Finally, unlike eco-technologies, environmental product innovations do not favor cost reduction. On the contrary, "they are more expensive" (E6) than a conventional product. Although they involve "an extra investment, it is what needs to be done and we do it" (E3). This suggests that environmental concern, embedded in the philosophy of these hotels, outweighs the high economic costs.

### 4.4. Social Aspects

Environmental process innovations satisfy the demand and have an impact on improving the business image of hotels by supporting their philosophy. In particular, foreign visitors "pay more attention to this issue" (E3) or "request using friendly systems in order to buy their packs" (E6). For certain hotels that seek "recognition from international organizations and certifications" (E3), environmental process innovations are mandatory requirements.

Similarly, environmental product innovations are associated with higher levels of guest satisfaction and a better corporate image, especially those with eco-labels. Although its implementation is low, there is an interest in incorporating TEIs in certified products or products with eco-labels.

"It backs us up (...), the customer can see the packaging and there is the certification" (E1)

"It is a plus to the given information and helps to convince the customer that the product is authentic" (E6)

Eco-labels are visible to guests, and therefore contribute to a positive perception of the quality of the accommodation service. Thus, they result in more suggestive and powerful communication elements than textual content, such as product specifications.

### 4.5. Barriers and Motivations

For environmental process innovations, investment is often the main constraint since they are not very affordable.

"You can have the intention to (...) try to be sustainable, (...), but it is expensive. [Environmental process innovations] help us (...) to sell more and that should help us implement more things, but sometimes it is not enough" (E3)

The previous testimony suggests that this heavy investment may be unlikely to be feasible for MSMEs in the sector. However, "if there is a long-term vision (...) there is a return on investment" (E2). In addition, by using environmental process innovations hotels add value to their service, so it is possible to increase the final price to compensate for costs, that is, "prices are readjusted according to the service offered" (E6). Another barrier identified by one of the interviewees was the lack of knowledge of the regulatory processes for the implementation of environmental systems in the public sector.

"This water (...) is treated with UV rays (...) and it is much cleaner than chlorinated water, but due to Peruvian regulations, we also must chlorinate it. [Moreover]

to use the wastewater treatment plant, you need a permit for implementation and for you to treat that sewage you need another permit (...) We went to the municipality, and they had no idea what we were talking about" (E5)

From the statement above, it is clear that current Peruvian regulations inhibit the correct use of eco-technologies and their potential environmental benefits.

For both TEIs, it was identified that there is a lack of providers in the areas where they work.

"We buy shampoo and soap in Lima. Here, in Oxapampa, it is very difficult to find providers of (...) green products. (...) In Lima, there are many more alternatives. So that is also (...) a challenge" (E2)

"Because of the location of the shelters, they all use biodigester tanks because they are not connected to the drainage network anywhere; they are installed in remote areas" (E3)

The location of hotels motivates the use of environmental process innovations as a response to the latent need to find alternatives to public utilities, but it limits the purchase of environmental product innovations.

## 5. Discussion and Conclusions

This study examined the importance of using TEIs from the perspective of Peruvian hotel managers. The results reveal that hotel managers deem eco-innovations to be important tools in their operations because of the various benefits that stem from them, especially when it comes to generating cost-savings and improving their corporate image [21,26,28,40–42].

The findings highlighted the use of CT and EOP in the rooms and maintenance areas, which is partially consistent with the study of Cuadros and Luza [34] where the room division implemented best environmental practices using TEIs. Although corrective innovations tend to be the most widely used [41], the results showed that participating hotels have adopted integrated and add-on innovations, understood as CT and EOP, respectively.

It was possible to identify that the main factors preventing a faster adoption of TEIs in these hotels were the shortage of suppliers, low financial capacity, and high prices. Notwithstanding, there is interest in progressively incorporating them into their operations [34,52]. In this regard, Rennings [2] suggests that at least one or more decades are required to consolidate the stages of invention, adaptation, and diffusion of TEIs in markets and different sectors. This explains the low levels of implementation in the hotel industry and the lack of suppliers in non-urban areas. In addition, in terms of economic barriers, goods and services are traditionally traded without reflecting true environmental costs. This market failure reinforces the preference for the adoption of conventional products over TEIs, which include economic and environmental costs, resulting in higher prices [24].

This research distinguished between environmental processes and product innovations and presented an independent analysis of each. Regarding innovations in environmental processes, it was found that those most used by hotels were those related to water and energy services, which coincides with previous studies [13–40]. Moreover, TEIs are considered as means that contribute to environmental responsibility, which coincides with the findings of Memarzadeh and Anand [48]. However, metrics should be used to determine the real impact of the use of environmental process innovations, especially in terms of reducing noise pollution, emissions, effluents, and waste [53]. In this regard, it is important to incorporate holistic assessment methods, such as life cycle assessment (LCA) and environmental indicators of consumption and performance [21,59,60].

On the economic level, it was found that even though environmental process innovations need large investment, these are implemented to reduce long-term production costs, be more eco-efficient, and add value to the service [24,26]. In some cases, the advances in TEIs have even allowed hotels to be more competitive, although there are discrepancies

in this regard [45,46,51]. In line with the above, Rennings and Zwick [60] suggest that the adoption of TEIs may be motivated by cost savings, but, in general, the motivation is often regulatory compliance. This differs from the current results, since, in Peru, legislation for the use of eco-innovations in the hotel industry is still incipient.

In relation to environmental product innovations, it was found that bathroom products were the most frequently adopted in these hotels. To a lesser extent, the use of eco-innovative products for cleaning public areas and laundry was observed. The lower use of environmental product innovations is due to the existence of few suppliers and higher prices than conventional alternatives. In addition, these TEIs do not reduce costs, but are used to meet demand and support the business philosophy and image. It was also noted that the decision to introduce environmental product innovations took into consideration their formulation with cleaner substances, but little attention was paid to other LCA principles (eco-efficiency or eco-design). The study by Ekinil et al. [61] confirms that hotel operations can be significantly affected by the use of eco-friendly products, so management should pay attention to their different characteristics, such as durability and potential hazards to air, soil, and water, among others. The similarities are also evident in the preference for products with minimal packaging and refillable packaging. The results are in line with those of other authors who recognize that the use of environmental product innovations implies higher costs, especially if they are certified, but motivates responsible environmental performance [44,47]. In addition, several authors have corroborated the need to improve the evaluation of TEIs based on multiple attributes, such as resource use, GHG and waste reduction, reuse and recycling, and eco-design [21,59]. However, cooperation with suppliers of these innovations should be encouraged [41]. Finally, demand preferences and the need for companies to maintain their environmental reputation were found to be pull factors for the adoption of ecological product innovations [2,25].

The significance of this research lies in the fact that it showed the environmental, economic, and social contributions of the use of TEIs in Peruvian hotel companies. It also revealed that TEI adoption is still slow, as various economic, commercial, and regulatory factors add to the fact that Peru's tourism superstructure is still unable to support and guide the hotel industry on its path to sustainability. Likewise, the concern of hotel owners and managers for the care of the environment must be complemented by actions to evaluate the effectiveness of the use of TEIs. Some of the citations denote a certain lack of interest in the actual environmental impact of their operations, either to further investigate the information provided by suppliers or to incorporate accurate environmental management metrics.

There were some limitations in this study. Peru's regulatory framework for the classification of lodging establishments only considers four types of businesses. Therefore, results may differ for other categories of accommodation establishments in other geographical contexts. Moreover, the qualitative data obtained are not statistically representative but they are relevant for the population studied, in this case the hospitality sector. Hence, we hope that future research can draw valuable lessons from this study. It is suggested to broaden the scope of analysis by including a multi-stakeholder perspective, such as guests and local communities, and connecting the study of TEIs to other topics of growing interest, such as certifications.

**Author Contributions:** Conceptualization, A.L.C.M. and G.F.M.V.; Methodology, A.L.C.M. and G.F.M.V.; Software, A.L.C.M. and G.F.M.V.; Validation, A.L.C.M. and G.F.M.V.; Formal analysis, A.L.C.M. and G.F.M.V.; Investigation, A.L.C.M. and G.F.M.V.; Resources, A.L.C.M. and G.F.M.V.; Data curation, A.L.C.M. and G.F.M.V.; Writing—original draft, A.L.C.M. and G.F.M.V.; Writing—review & editing, A.L.C.M., G.F.M.V. and D.Á.B.; Visualization, A.L.C.M. and G.F.M.V.; Supervision, A.L.C.M., G.F.M.V. and D.Á.B.; Project administration, A.L.C.M. and G.F.M.V. All authors have read and agreed to the published version of the manuscript.

**Funding:** This research received no external funding. The APC was funded by Universidad San Ignacio de Loyola.

**Institutional Review Board Statement:** Not applicable.

**Informed Consent Statement:** Informed consent was obtained from all subjects involved in the study.

**Data Availability Statement:** No new data were created or analyzed in this study. Data sharing is not applicable to this article.

**Conflicts of Interest:** The authors declare no conflict of interest.

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
