# Peer review of "Analysis of Eco-Innovations in Peruvian Accommodation Establishments"

_sustainability, doi:10.3390/su15086700_

Round 1
Reviewer 1 Report
In the abstract of the authors, a lot of attention is paid to the relevance of the issue, which does not cause problems. But we do not understand what the purpose of the research is, what the authors are trying to achieve with these studies, what methods and models they used for this and what the overall result was. Adding these elements to the abstract will enable readers to understand not only the relevance of the study, but also its significance.
There are questions about the survey methodology itself. The authors conducted a survey of 60 hotel guests, they are familiar with the question of how representative such a sample is? "Note: The sample consisted of three international guests and fifty-seven domestic guests." Also, the authors do not provide a margin of error for this survey.
It should be noted that the general conclusions are more descriptive and have some overlap with the text of the article. But, the positive thing is that they provide identification of the problem and obtained research results. The conclusion of the conclusions "Hence, it is recom- 359
corrected to exercise caution in generalizing and drawing conclusions based on results. Despite these 360
limitations, it is expected that future research will be able to learn valuable lessons from 361
this research.”
Author Response
Point 1: In the abstract of the authors, a lot of attention is paid to the relevance of the issue, which does not cause problems. But we do not understand what the purpose of the research is, what the authors are trying to achieve with these studies, what methods and models they used for this and what the overall result was. Adding these elements to the abstract will enable readers to understand not only the relevance of the study, but also its significance.
Response 1: Thank you for your comments. We've improved the abstract by making various adjustments to the chapters. We are now only including the qualitative results. Thus, the methodology and results sections will only relate to the qualitative part of the study.
Point 2: There are questions about the survey methodology itself. The authors conducted a survey of 60 hotel guests, they are familiar with the question of how representative such a sample is? "Note: The sample consisted of three international guests and fifty-seven domestic guests." Also, the authors do not provide a margin of error for this survey.
Response 2: Because the sample size was small and can't be considered representative enough, we have now taken out the quantitative results from the paper. Therefore, the methodology and results related to this aspect have been modified.
Point 3: It should be noted that the general conclusions are more descriptive and have some overlap with the text of the article. But, the positive thing is that they provide identification of the problem and obtained research results. The conclusion of the conclusions "Hence, it is recom- 359
corrected to exercise caution in generalizing and drawing conclusions based on results. Despite these 360
limitations, it is expected that future research will be able to learn valuable lessons from 361
this research.”
Response 3: As stated before, the quantitative results were the ones that were most affected by the limitations. Thus, we decided to take them out altogether to avoid making generalizations from the takeaways of this research.
Reviewer 2 Report
The authors took up an extremely interesting and current topic of pro-ecological innovations in tourist accommodation facilities. Unfortunately, I have the impression that the methodological shortcomings in this case are so significant that this article needs to be modified.
The main problem here is focusing on ecolodge facilities, which for many reasons are incomparable to other accommodation facilities - e.g. with large hotels. These issues are discussed in the commentary in the text. In general, the type of these objects was not discussed in the article, which is a big mistake. Comparing the results of research conducted mainly among guests of ecolodges with the results of research conducted in large hotels is a misunderstanding and should not take place in scientific research published in an IF journal.
Moreover, the number of respondents is definitely too small for the purposes of a quantitative study. In addition, we do not know how many of these guests were in ecolodges and how many in other facilities.
These and other comments are discussed in the text of the article.
If this research was carried out only among ecolodges guests, it would make sense and value, but in this case it is not. This needs to be corrected/modified. It is highly recommended (and even necessary) to increase the number of respondents.

Author Response
Point 1: The authors took up an extremely interesting and current topic of pro-ecological innovations in tourist accommodation facilities. Unfortunately, I have the impression that the methodological shortcomings in this case are so significant that this article needs to be modified.
Response 1: We have modified the methodology section according to your and other revisors' suggestions. We are now only focusing on the qualitative section of this research.
Point 2: The main problem here is focusing on ecolodge facilities, which for many reasons are incomparable to other accommodation facilities - e.g. with large hotels. These issues are discussed in the commentary in the text. In general, the type of these objects was not discussed in the article, which is a big mistake. Comparing the results of research conducted mainly among guests of ecolodges with the results of research conducted in large hotels is a misunderstanding and should not take place in scientific research published in an IF journal.
Response 3: We hadn't previously specified this aspect but we have now included more clarification on the type of accommodation establishments that were part of this study. All of the participating hotels in this study are hotels. We initially included the term ecolodge because most of these businesses self-proclaimed themselves to be one. However, these establishments weren't necessarily eco-lodge in the true sense of the word, meaning they aren't small or family-owned businesses. This term was rather being used as a way for businesses to highlight certain attributes that differentiated them from other establishments (e.g. being more environmentally friendly than the majority of accomodation establishments). Additionally, Peruvian legislation doesn't recognize ecolodges in their classification.
Point 4: Moreover, the number of respondents is definitely too small for the purposes of a quantitative study. In addition, we do not know how many of these guests were in ecolodges and how many in other facilities.
These and other comments are discussed in the text of the article.
Response 4: As stated above, we have modified the study to only focus on the qualitative part.
Point 5: If this research was carried out only among ecolodges guests, it would make sense and value, but in this case it is not. This needs to be corrected/modified. It is highly recommended (and even necessary) to increase the number of respondents.
Response 5: This research originally included guests from the 6 participating hotels. However, as stated before, we've taken out the quantitative section.
You can find a response to each of your comments in the attached document.

Reviewer 3 Report
The authors are commended for presenting a study that could add to sustainable practices in the hospitality industry. The importance of TEIs is well elaborated and authors used relevant literature. However, this study could be improved by:
1. Ensuring that there is alignment in the purpose of the study in the ab7stract and the main document. One or more research questions could assist authors to stay focused on the purpose of the study.
2. Methodology - Providing full details of the research design for both the quantitative and qualitative study. Though the authors stated the instrument used in the data collection, they did not specify details such as how they were administered, duration of the interviews, how participants response were captured etc. The authors did not provide details how the data were analyzed. Furthermore, the authors did not discuss how they ensured validity and reliability in the quantitative and rigor in the qualitative study. Consider more details in the attached document.
3. Results - The quantitative results are analyzed in terms of frequency and percentage, which is too general and do not provide in depth analysis of the results. Authors could consider use descriptive statics such as measure of central tendency and/or measures of variability.
For the qualitative study: How did the authors reach to these findings? What method of analysis did they applied? Could they have used thematic analysis, then they should give full details in the methodology section.

Author Response
Point 1: Ensuring that there is alignment in the purpose of the study in the abstract and the main document. One or more research questions could assist authors to stay focused on the purpose of the study.
Response 1: Thank you for your comments. We have improved the summary by limiting our objective to analyzing the importance of TEIs in Peruvian lodging establishments from the perspective of hotel managers. If this was not clear enough previously, we have now made sure that it is evident since we have chosen to work on this research only under the qualitative approach.
Point 2: Methodology - Providing full details of the research design for both the quantitative and qualitative study. Though the authors stated the instrument used in the data collection, they did not specify details such as how they were administered, the duration of the interviews, how participants’ responses were captured, etc. The authors did not provide details on how the data were analyzed. Furthermore, the authors did not discuss how they ensured validity and reliability in the quantitative and rigor in the qualitative study. Consider more details in the attached document.
Response 2: Because the sample size was small and can't be considered representative enough, we have now taken out the quantitative results from the paper. In the qualitative part, we have provided more information on the technique, tool, and duration of the application of the same.
Point 3. Results: Authors could consider using descriptive statics such as measures of central tendency and/or measures of variability. For the qualitative study: How did the authors reach these findings? What method of analysis did they apply? Could they had used thematic analysis, then they should give full details in the methodology section. Limitations, it is expected that future research will be able to learn valuable lessons from 361 this research."
Response 3: As stated before, the quantitative results were the ones that were most affected by the limitations. Thus, we decided to take them out altogether to avoid making generalizations from the takeaways of this research. On the other hand, modifications have been made in the methodology section and now the methodology used for the analysis has been clearly described from a qualitative approach (technique, tools, and analysis method).
We are attaching a more detailed response to each one of the comments made on the manuscript.

Round 2
Reviewer 2 Report
Accept in present form.
Author Response
Thank you for your suggestions. We have made some minor modifications to the document following your and other reviewers' advice.
Reviewer 3 Report
Dear Authors
Thank you very much for taking time to review your work. You are greatly commended for all the effort you have put.
There are few comments I have included that could improve you work. See comments in the attached document.

Author Response
Dear reviewer,
Thank you for your valuable comments.
We've addressed your observations regarding the methodology section. We took out the mention of in-depth interviews and we included how we ensured rigor. We also fixed the format mistakes.